# Preoperative Tumor Texture Analysis on MRI for High-Risk Disease Prediction in Endometrial Cancer: A Hypothesis-Generating Study

**DOI:** 10.3390/jpm12111854

**Published:** 2022-11-07

**Authors:** Maura Miccò, Benedetta Gui, Luca Russo, Luca Boldrini, Jacopo Lenkowicz, Stefania Cicogna, Francesco Cosentino, Gennaro Restaino, Giacomo Avesani, Camilla Panico, Francesca Moro, Francesca Ciccarone, Gabriella Macchia, Vincenzo Valentini, Giovanni Scambia, Riccardo Manfredi, Francesco Fanfani

**Affiliations:** 1Area Diagnostica per Immagini, Dipartimento Diagnostica per Immagini, Radioterapia Oncologica ed Ematologia, Fondazione Policlinico Universitario A. Gemelli, IRCCS, 00168 Rome, Italy; 2U.O.C. Radioterapia Oncologica, Dipartimento Diagnostica per Immagini, Radioterapia Oncologica ed Ematologia, Fondazione Policlinico Universitario A. Gemelli, IRCCS, 00168 Rome, Italy; 3Department of Obstetrics and Gynaecology, Institute for Maternal and Child Health, IRCCS ‘Burlo Garofolo’, 34137 Trieste, Italy; 4Gynecologic Oncology, Gemelli Molise Spa, 86100 Campobasso, Italy; 5Radiology Department, Gemelli Molise Spa, 86100 Campobasso, Italy; 6Dipartimento Scienze della Salute della Donna, del Bambino e di Sanità Pubblica Fondazione Policlinico Universitario A. Gemelli, IRCCS, 00168 Rome, Italy; 7Radiotherapy Unit, Gemelli Molise Hospital, 86100 Campobasso, Italy; 8Sede di Roma, Università Cattolica del Sacro Cuore, 00168 Rome, Italy; 9Istituto di Clinica Ostetrica e Ginecologica, Università Cattolica del Sacro Cuore, 00168 Rome, Italy

**Keywords:** radiomics, endometrial cancer, magnetic resonance imaging

## Abstract

Objective: To develop and validate magnetic resonance (MR) imaging-based radiomics models for high-risk endometrial cancer (EC) prediction preoperatively, to be able to estimate deep myometrial invasion (DMI) and lymphovascular space invasion (LVSI), and to discriminate between low-risk and other categories of risk as proposed by ESGO/ESTRO/ESP (European Society of Gynaecological Oncology—European Society for Radiotherapy & Oncology and European Society of Pathology) guidelines. Methods: This retrospective study included 96 women with EC who underwent 1.5-T MR imaging before surgical staging between April 2009 and May 2019 in two referral centers divided into training (T = 73) and validation cohorts (V = 23). Radiomics features were extracted using the MODDICOM library with manual delineation of whole-tumor volume on MR images (axial T2-weighted). Diagnostic performances of radiomic models were evaluated by area under the receiver operating characteristic (ROC) curve in training (AUCT) and validation (AUCV) cohorts by using a subset of the most relevant texture features tested individually in univariate analysis using Wilcoxon–Mann–Whitney. Results: A total of 228 radiomics features were extracted and ultimately limited to 38 for DMI, 29 for LVSI, and 15 for risk-classes prediction for logistic radiomic modeling. Whole-tumor radiomic models yielded an AUCT/AUCV of 0.85/0.68 in DMI estimation, 0.92/0.81 in LVSI prediction, and 0.84/0.76 for differentiating low-risk vs other risk classes (intermediate/high-intermediate/high). Conclusion: MRI-based radiomics has great potential in developing advanced prognostication in EC.

## 1. Introduction

Medical imaging plays an essential role in clinical management of cancer patients to support diagnosis or to assess a clinical stage. Until recently, the traditional practice of radiology was centered largely on subjective visual interpretation by radiologists based on their training and experience. However, in this emerging era of precision medicine, the mere description of tumor extent may be insufficient when providing more objective clinically relevant information. Advances in high-throughput computing have facilitated the development of processes for conversion of biomedical images into robust and validated biomarkers; this practice is termed “radiomics” [1]. Radiomics is an advanced image analysis technique that converts diagnostic images into quantitative data, extracting features from specific regions of interest of selected volumes. These features can be correlated with clinical or histopathological factors, allowing a noninvasive characterization of tumors and offering information about underlying tumor heterogeneity and aggressiveness.

Endometrial cancer (EC) is the most common gynecological malignancy in industrialized countries and represents the sixth-most diagnosed cancer in women [2]. Conventionally, postoperative clinicopathologic findings such as tumor histology, stage of disease according to International Federation of Gynecology and Obstetrics (FIGO) [3], grade of histological differentiation, and lymphovascular space invasion (LVSI) are the key prognostic factors in EC. A risk classification including LVSI was proposed in 2016 [4]. Recently, the ESGO-ESTRO-ESP (European Society of Gynaecological Oncology—European Society for Radiotherapy & Oncology and European Society of Pathology) published updated guidelines for risk group determination in EC, integrating both clinicopathologic variables and molecular diagnostics [5], classifying patients into five risk classes (low risk, intermediate risk, high-intermediate risk, high risk, advanced/metastatic). Different surgical and adjuvant therapeutic strategies have been recommended for these different risk groups [5,6,7]. Therefore, the correct placing of EC within this prognostic stratification framework allows the most appropriate and individualized treatment. Magnetic resonance imaging (MRI) has a pivotal role in pretreatment assessment of EC, being highly specific in evaluation of deep myometrial invasion (DMI), cervical stromal involvement, and lymph node metastasis [8]. Recently, radiomic tumor profiling based on MRI has been proposed as a tool for accurate diagnosis, preoperative risk stratification, or assessment of treatment response in several cancer types, such as cervical cancers [9,10,11]. Despite promising performances of this field in clinical practice, only a few studies have explored MRI-based radiomic tumor features in endometrioid EC and linked these to an aggressive phenotype [12,13,14]. Thus, the aim of this study is to create and validate a radiomics model based on staging MRI in patients affected by EC for the prediction of DMI and LVSI, which to date represent the most relevant histopathological prognostic factors in decision-making on adjuvant therapy in early stages of EC. We also developed and validated a radiomics MRI-based model able to differentiate low-risk EC from the other ECs, as proposed by the ESGO/ESTRO/ESP evidence-based guidelines [5].

## 2. Materials and Methods

### 2.1. Study Design and Population Selection

This is a two-center retrospective observational study. The Ethical Committee of reference approved the study. The study population included women diagnosed with EC and surgically treated at Fondazione Policlinico Universitario “A. Gemelli” IRCCS of Rome, Italy (Center 1) and at Gemelli Molise Spa, Campobasso, Italy (Center 2) from April 2009 to May 2019. Inclusion criteria were: (i) patients with histological diagnosis of EC and (ii) patients who had a preoperative staging 1.5-T MRI and availability of digital images in DICOM format. Exclusion criteria were as follows: (i) uterine carcinosarcomas, atypical hyperplasia, ambiguous histology (such as synchronous cervical and endometrial cancer) and (ii) patients without available MR images or with images deemed of insufficient quality to visualize the endometrial lesion (e.g., artefacts). Patients recruited in Center 1 were included as the training set, whereas patients enrolled in Center 2 were classified as the external validation cohort. Patients of Center 1 were retrospectively identified through specific queries on institutional RedCap databases (electronic data capture tools hosted at https://redcap-irccs.policlinicogemelli.it/ (accessed on 1 June 2019)), whereas patients of Center 2 were retrospectively identified from the database of the local department of gynecological oncology. Preoperative staging pelvic 1.5-T MRI exams (available in digital form and stored in the institutional PACS systems) of patients who had been selected for study were screened for quality assurance by an expert radiologist to be selected for radiomics analysis. All the included patients underwent surgery in local institutions by trained gynecologic oncologic surgeons, and the applied surgical approach was based on both patient and disease characteristics. Surgical specimens were assessed by a dedicated oncologist pathologist in each institution, and the presence of DMI (≥50%), LVSl, cervical stromal invasion, lymph node metastases, and tumor histologic subtyping and grading were confirmed microscopically. Clinical data, histopathological parameters and FIGO stage were collected. For patients selected from Center 1, information was obtained from Redcap, whereas for patients selected from Center 2, information was obtained from patients’ medical records. According to ESGO/ESTRO/ESP guidelines [5], low-risk EC was defined as endometrioid EC with myometrial invasion <50%, tumor grades 1–2, negative or focal LVSI, and no extrauterine invasion. The definition of prognostic risk groups was obtained even if the molecular classification was unknown, as the interval of the recruitment time did not guarantee availability of the molecular assay for the study populations.

### 2.2. MR Imaging

In both centers, all MRI imaging studies were acquired according to local institutional diagnostic protocols and consisted of a 1.5 T MR scan (Signa Excite; GE Healthcare, Little Chalfont, United Kingdom) using an eight-element pelvic phased-array surface coil. The pelvic MR imaging protocol in the training and validation sets included T2-weighted imaging (WI) in the axial, sagittal, and axial oblique planes; Diffusion Weighted Imaging (DWI); and T1-W postcontrast images (Table 1).

### 2.3. Image Analysis 

The endometrial primary tumors were manually delineated using the free, open-source software application ITKSNAP (www.itksnap.org, v. 3.8.0 (accessed on 1 July 2019)). The segmentations were performed by two radiologists with expertise in gynecological imaging (having 10 and 4 years of experience in pelvic MRI reading, respectively), and the cases were randomly assigned between the two operators. The radiologists, who were aware of the diagnosis of endometrial cancer but blinded to clinical and pathologic outcomes of patients, manually segmented tumor regions of interest (ROI) for each case. An ROI was manually drawn along the margin of the visible gross tumor on the axial plane T2-WI along the tumor boundary on each slice for each tumor (Figure 1). 

The ROIs were manually delineated along the lesion boundaries, obtaining whole-tumor data. Tumor contour was defined as the area of intermediate signal intensity on T2-WI that was different from normal adjacent low-signal-intensity myometrium (Figure 2). Although not used for segmentation, DWI- and T1-WI-acquired postcontrast injection series were available to the readers for visual inspection support segmentations in case they were needed. The segmented images were then exported to MODDICOM, an R library specifically designed to perform radiomics analysis [15]. To remove any eventual bias due to the different spatial resolution, all the images were resampled to a mean value of spatial planar resolution equal to 0.548 × 0.548 mm^2^ before features extraction. All radiomics features were extracted using MODDICOM, fully compliant with the Image Biomarker Standardization Initiative recommendations [16]. The original MRI file and the corresponding ROI segmentation masks were simultaneously uploaded and double checked by an independent operator to ensure anatomical consistency. The following families of features were taken into account: intensity-based statistical, morphological, and textural features (grey-level co-occurrence features, run length features and size zone features). 

### 2.4. Statistical Analysis

Sample size calculation was based on the primary objective. Of the different objectives of this study, this one requires the largest number of patients to ensure the stability of the prediction model. Simulation studies such as Peduzzi et al. [17] demonstrated that logistic regression models require 12–15 events per predictor to produce stable estimates, as confirmed also by the TRIPOD statements [18]. The analysis was performed after segmentations test/retest, aiming to assess features reproducibility. The radiomics features were extracted also from Laplacian of Gaussian (LoG) and wavelet-filtered images for noise removal and pattern enhancing. The radiomics analysis methods are reported in details hereafter: (1)Features reproducibility and univariate analysis. For the univariate analysis, the features space dimensionality was reduced by a Pearson correlation test by setting a correlation threshold of 0.90 in order to remove collinear features. In addition to features correlation analysis, features reproducibility analysis was carried out in order to identify features strongly dependent on slight variations in the contouring and thus less reproducible. This features reproducibility assessment was done via intraclass correlation coefficient (ICC) analysis. Based on the 95% confidence interval of the ICC estimate, values less than 0.5, between 0.5 and 0.75, between 0.75 and 0.9, and greater than 0.90 are indicative of poor, moderate, good, and excellent reliability, respectively. Features with poor reliability were excluded from the subsequent steps of the analysis. After having selected the contour-stable features and reduced the dimensionality of feature space, the remaining features were tested for association with the outcome using the Wilcoxon–Mann–Whitney test against the considered binary outcome (i.e., MI > 50%; LVSI positive; low risk class), setting the significance level to a *p*-value < 0.05. The number of significant test results was then compared to the expected number of type I errors to account for multiple testing [19].(2)Features outcome and predictive models. Logistic regression models were trained and tested to identify radiomics signatures able to predict each considered clinical outcome. The models were trained starting from features significant at the univariate analysis and further refined through cross-validation AIC-based stepwise selection and logistic least absolute shrinkage and selection operator (LASSO) selection, with respect to each considered clinical outcome variable. The area under the curve (AUC) of the receiver operator characteristic (ROC) was used to evaluate the predictive accuracy of the radiomics models developed. The sample performance metrics were estimated from cross-validation ROC AUC curves and classification matrix statistics on the testing set (sensitivity, specificity, positive predictive value, negative predictive value, accuracy) and assessed on the external dataset. For the LASSO radiomic signatures derived in the training cohort, optimal cutoffs were identified from the ROC curves using the Youden Index.

## 3. Results

Ninety-eight patients examined at Center 1 were identified for the training set. An overview of patients’ characteristics is reported in Table 2. To identify predictive models to be used in the clinical setting, we decided to consider exclusively the endometrioid histotype for the analysis. We excluded serous/clear cell histotypes (19 patients) as they are considered high-grade tumors regardless of other prognostic factors and are already categorized in high-risk classes. For the final analysis, two patients with metastatic EC were excluded due their overall low number. Four patients had low-quality MR images and were excluded from radiomic analysis due to technical problems. A total of 73 patients was finally considered for the training set (Figure 3). Twenty-six patients examined at Center 2 were selected for the external validation set (Table 2). Among them, 23 had available digital MR images. Surgical staging included hysterectomy with bilateral salpingo-oophorectomy for all the patients; lymph nodal assessment includes pelvic lymphadenectomy and accompanying paraaortic lymphadenectomy or sentinel lymph node.

For the training set, postoperative histologic assessment revealed myometrial invasion <50% in 36 patients (49%), myometrial invasion ≥50% in 37 patients (51%); for the validation set, myometrial invasion <50% was observed in 14 patients (61%), while myometrial invasion ≥50% was reported in 9 patients (39%). Among 73 patients of the training set, 53 of them had their LVSI status with LVSI negative in 26 patients (49%) and LVSI positive in 27 patients (51%). Among 23 patients of the validation set, LVSI was available in 18 patients with LVSI negative in 12 patients (67%) and LVSI positive in 6 patients (33%). 

Applying the 2020 ESGO/ESTRO/ESP risk assessment system with unknown molecular classification [5], the 73 patients of the training set presented the following distribution for classes of risk: 18 (25%) low risk and 55 (75%) intermediate/high-intermediate/high risk. The 23 patients of the validation set presented the following classes of risk distribution: 10 (43%) low risk and 13 (56%) intermediate/high-intermediate/high risk.

A total of 228 radiomics features were extracted from T2-W MR images. The features selected during the training phase were used for the final radiomics analysis, and they were 38 for the DMI dataset, 29 for the LVSI dataset, and 15 for the risk class dataset. The most significant radiomics feature for the prediction of DMI was “F_ F_cm.info.corr.1” (*p* = 0.0003). The most significant radiomics feature for the prediction of the presence of LVSI was “F_cm.info.corr.1” (*p* = 0.01). The most significant radiomics feature for the prediction of the risk class in the cohort was “F_szm_2.5D.szlge” for the low risk (*p* = 0.01) (Figure 4). 

In prediction of DMI, the performance was AUC 0.85 on the internal cohort. The model showed the following values: Sensitivity 0.67, Specificity 0.89, Positive Predictive Value (PPV) 0.86, Negative Predictive Value (NPV) 0.72, and Accuracy 0.78 [95% CI 0.67–0.86]. Optimal Cut-Point according to Youden index was 0.69. In the validation set, the radiomics model for predicting DMI showed an AUC value of 0.68 with Accuracy of 0.69 [95% CI 0.47–0.86], Sensitivity of 0.66, Specificity of 0.71, PPV of 0.60, and NPV of 0.76 (Table 3).

Considering the full dataset, the radiomics model achieved the best diagnostic performance for prediction of LVSI, with AUC 0.925 on the internal cohort. In estimation of LVSI, the radiomics model also showed the highest value of Sensitivity (1.00) and an NPV of 1.00, with Specificity, PPV, and Accuracy of 0.77, 0.81, and 0.89 [95% CI 0.77–0.95], respectively. Optimal Cut-Point according to Youden index was 0.35. In the validation set, the radiomics model for predicting LVSI achieved AUC 0.81, with Accuracy of 0.83 [95% CI 0.58–0.96], Sensitivity of 0.83, Specificity of 0.83, PPV of 0.71, and NPV of 0.90 (Table 3).

In the training set, the radiomics model for predicting the low-risk EC group (stage I, MI < 50%, LVSI negative, endometrioid histotype) vs the other risk groups (intermediate/high-intermediate/high risk) showed AUC 0.84. The model showed a risk prediction of low-grade EC with a value of Sensitivity 0.64, Specificity 0.93, PPV 0.73, NPV 0.89, and Accuracy 0.86 [95% CI 0.76–0.93]. Optimal Cut-Point according to the Youden index was 0. 41. The radiomics model to predict low-risk showed AUC 0.76 in the validation set with Accuracy of 0.82 [95% CI 0.61–0.95], Sensitivity of 0.60, Specificity of 1, PPV of 1, and NPV of 0.76 (Table 3).

The ROC curves for all the models in training and in validation sets are shown in Figure 5.

## 4. Discussion

This study showed that MRI-based whole-tumor radiomics analysis yielded medium-to-high diagnostic performance for prediction of high-risk surgico-pathological features in EC, as the presence of DMI and LVSI. Furthermore, we explored the capability of MRI-based radiomics for preoperative prediction of risk class in EC. The radiomics model developed in our study showed a moderate-to-good ability to discriminate between low-risk EC and the other classes, with promising reproducibility and reliability, as confirmed by the external validation process (TRIPOD 3). 

The challenge in treatment planning and prognostication in EC is the preoperative assessment of risk factors tailoring surgery and subsequent therapy, such as DMI, LVSI and nodal metastasis. In the evaluation of DMI, standard MRI showed high values of sensitivity and specificity ranging from 81% to 90% and from 82% to 89%, respectively [20]. However, standard qualitative MRI evaluation seems to be strongly dependent on reader experience, with a relatively high interobserver variability [21]. The radiomics model developed in this study might be helpful in identification of DMI, being consistent with preliminary studies in EC [12,22,23,24]. Ueno et al. reached an accuracy of 81% in identification of DMI, employing eleven features derived from T2-WI, DWI, ADC, and postcontrast images from pelvic MR scans of 137 patients who underwent surgery for endometrial cancers [12]. Ytre-Hauge and colleagues obtained an accuracy of 78% for DMI detection with a single texture feature derived from ADC maps [23]. Both reported studies extracted radiomic features from primary tumor manual segmentations in a single image plane. Ytre-Hauge et al., however, did not report radiomic modeling and neither Ueno et al. nor Ytre-Hauge et al. validated their findings on an external validation cohort. In a larger study performed on whole tumor segmentations on postcontrast T1-WI, the authors found a moderate accuracy for predicting DMI, both in the training (AUC 0.84) and the validation sets (AUC 0.74) [25]. In a recent study of 54 patients affected by EC, Stanzione et al. found that their random forest-based radiomic model was able to predict DMI with an AUC of 0.92 and 0.94 in the training and validation sets, respectively [26], concluding that the radiomics model could increase radiologists’ performance in interpreting correctly DMI. Despite being based on different approaches in cohort sizes, imaging sequences, radiomic data extraction, and statistical methods (Table 4) [27], all the reported studies and our results confirm a promising role of MRI-based radiomics features as an adjunct tool to the standard MRI evaluation for DMI, offering clinical benefit in challenging cases, such as anatomic uterine distortions, leiomyomas, presence of adenomyosis, or small endometrial tumor. We chose to focus exclusively on T2-WI because T2-WI is an essential component of pelvic MRI and provides high spatial resolution and tissue-specific contrast, when compared to DW and DCE imaging. Texture parameters reflect pixel heterogeneity of the T2-W images, which are influenced by many parameters, including neoplastic cellular infiltration, cellular and interstitial oedema, and blood vessel density and distribution. We showed that quantitative T2-W image features have the potential to serve as noninvasive markers for assessing aggressiveness in EC.

LVSI is the single prognostic factor that cannot be preoperatively detected with conventional diagnostic tools, including MRI and endometrial biopsy. Besides being a negative prognostic risk factor related to reduced PFS and OS, especially for early-stage endometrial tumors, the evaluation of presence or absence of LVSI is even more necessary to guide adjuvant therapy in case of inadequate surgical lymph node staging. Prior studies [12,29,31] in which two- and three-dimensional MRI-based features were used reported lower performance for LVSI prediction. When compared to these previous experiences (Table 4), our study demonstrated excellent diagnostic accuracy of the radiomic LVSI prediction model. In particular, 29 features were strongly correlated with LVSI with AUC 0.92 and with a sensitivity of 1.00 and negative predictive value of 1.00. Considering the cut-point by Youden index, our radiomics model misclassified only 6 out of 33 patients to positive LVSI in the training cohort. Notably, our radiomic signatures based on whole-tumor MRI radiomics yielded similar performance metrics in the validation cohort, suggesting the generalizability of whole-tumor radiomic profiling in EC. Nevertheless, despite the methodological robustness of the analysis, our model still has the potential risk of a certain degree of overfitting, since we included 53 patients in the final analysis and further validation on larger cohorts is needed to confirm these observations.

High-risk patients have endometrial tumors with at least one of the following characteristics: DMI, high-grade tumor, non-endometrioid histological subtype (serous and clear cell), LVSI, extrauterine spread, or nodal involvement. Prognostic stratification based exclusively on histopathological characteristics is still widely used in many centers where molecular classification is not yet available [5]. Furthermore, molecular features might not be available until after hysterectomy. High- and low-risk EC are the two opposite groups than could be considered for different surgical treatments. Current international guidelines recommended that patients with high-risk EC should be treated with total hysterectomy, bilateral salpingo-oophorectomy (THBSO), lymphadenectomy (LA), or adjuvant therapy. Conversely, LA could be omitted in patients with low-risk EC in which THBSO is the standard treatment [5]. A recent report developed an MRI- and clinical-based radiomics nomogram model by combining whole-volume radiomics features extracted from multiparametric MRI and clinical parameters in a large multicenter dataset of patients with EC, with the aim to predict high-risk patients (referring to EC needing lymphadenectomy) [13]. This nomogram achieved good diagnostic performance with an AUC of 0.896 and good net benefit by clinical decision curve analysis for high-risk EC [13]. The ESGO/ESMO/ESP classification is one of the most routinely used classifications to predict lymph node invasion and thus to optimize surgical planning. In prediction of ESGO/ESMO/ESP risk groups [5], our radiomics model obtained good results, with an accuracy of 0.86, a specificity of 0.93 and negative predictive value of 0.89 in the prediction of low-risk EC. Since radiomics can provide information regarding preoperative risk stratification, standard preoperative MR images radiomics-based models could be useful in EC to achieve optimal selection of patients, avoiding overtreatment in low-risk disease [32]. Our findings deserve further investigation, aiming to increase their performance and to better describe their possible translational application in daily clinical practice. 

This study has nevertheless some limitations. First, its retrospective nature represents an inevitable source of selection bias and imaging data inhomogeneity. Data were collected over a long time (2009–2019), and MRI technology has improved with more sophisticated machinery, different technical parameters, and better image quality. All these aspects might have impacted the textural radiomic features extraction and subsequent modeling. Second, the number of patients was relatively low, even if it allowed for a dataset for further model validation. This also limited the statistical power of testing to assess differences in performance. Whole-tumor segmentations were manually delineated instead of being semiautomatically/automatically segmented, thus making it difficult to avoid subjective errors, even if segmentation-features dependency has been explored [34,35]. Finally, we focused only on radiomics features extracted from T2-WI, supported by recent research [36]. The effect of other routine sequences, such as DWI and contrast-enhanced MRI, were not investigated. However, the integration of this information may represent a great challenge and it will be taken into account to improve the power of our model in subsequent investigations. 

## 5. Conclusions

In conclusion, we developed radiomics-based predictive models with encouraging performances to identify prognostic factors conditioning surgical and adjuvant therapy, such as DMI and LVSI, the latter not detectable before surgical staging. The same models applied to the risk classes conferred a good performance in the prediction of preoperative risk, helping to stratify patients prior to surgery. The next step could be the association of radiomic and genomic in the radiogenomic analysis, with the goal to associate radiomic features with the tumor genomic profiling that has recently been introduced to differentiate endometrial cancer patients into four classes [37].

We believe that radiomics is a field worth continuing to explore in patients with endometrial cancer, representing a reproducible diagnostic support tool, although further investigations are necessary to evaluate its performances before its effective release in clinical practice.

## Figures and Tables

**Figure 1 jpm-12-01854-f001:**
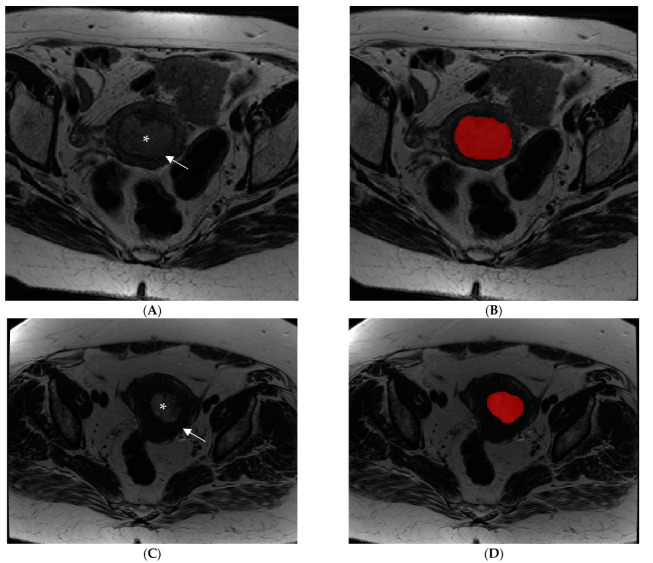
Examples of manual segmentation of endometrial tumors observed by MRI using an Axial Fast Spin Eco T2-weighted image acquisition sequence. (**A**,**B**) Images show an intermediate signal intensity lesion (asterisk) causing more than 50% myometrial invasion (arrow) in a patient with high-risk endometrial cancer before (**A**) and after (**B**) image segmentation. (**C**,**D**) Images show a patient with a low-risk endometrial tumor (asterisk) with less than 50% myometrial invasion (arrow) before (**C**) and after (**D**) image segmentation.

**Figure 2 jpm-12-01854-f002:**
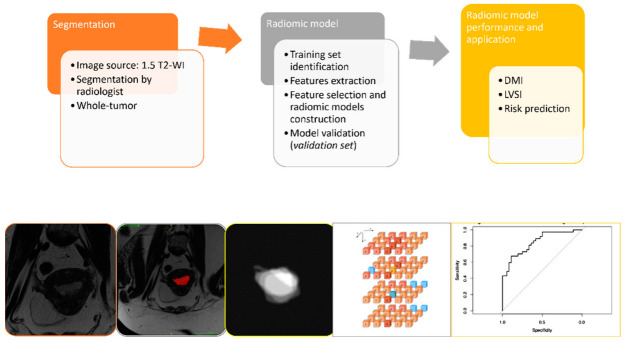
Outline of the project workflow consisting of whole-volume manual tumor segmentation on T2-weighed images, radiomic tumor feature extraction, and construction of radiomic signatures for prediction of selected outcomes in EC patients. The radiologist defined ROIs on T2-WI to extract intratumoral radiomic features; radiomic signatures were derived based on the whole-tumor masks (whole-tumor radiomics). Least absolute shrinkage and selection operator (LASSO) was applied for prediction modeling.

**Figure 3 jpm-12-01854-f003:**
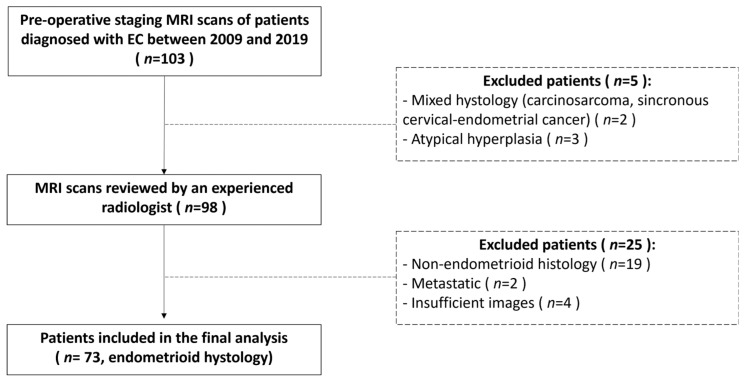
Flow chart describing patients’ selection process and exclusion criteria for the training set.

**Figure 4 jpm-12-01854-f004:**
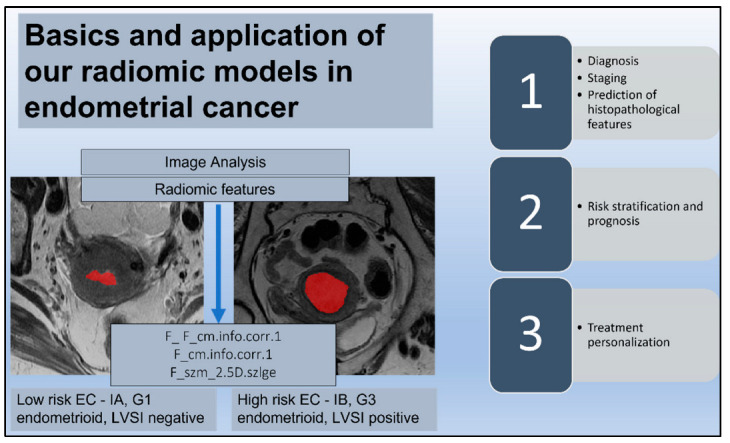
Radiomic signature. The endometrial tumor is delineated by a red area on T2-weighted images in two different patients with a histological diagnosis of endometrial cancer (EC); low risk ECs are defined as stage IA endometrioid, grade (G) 1–2, and lymph vascular space invasion (LVSI) negative; high-intermediate risk ECs are identified as stage IB endometrioid, G 3, regardless of LVSI status. The reported radiomic features (F) are the most significant for the prediction of histopathological factors such as deep myometrial invasion and LVSI and for the discrimination of low-risk ECs. The diagram also shows possible correlations and applications of radiomic signatures and clinical factors in EC.

**Figure 5 jpm-12-01854-f005:**
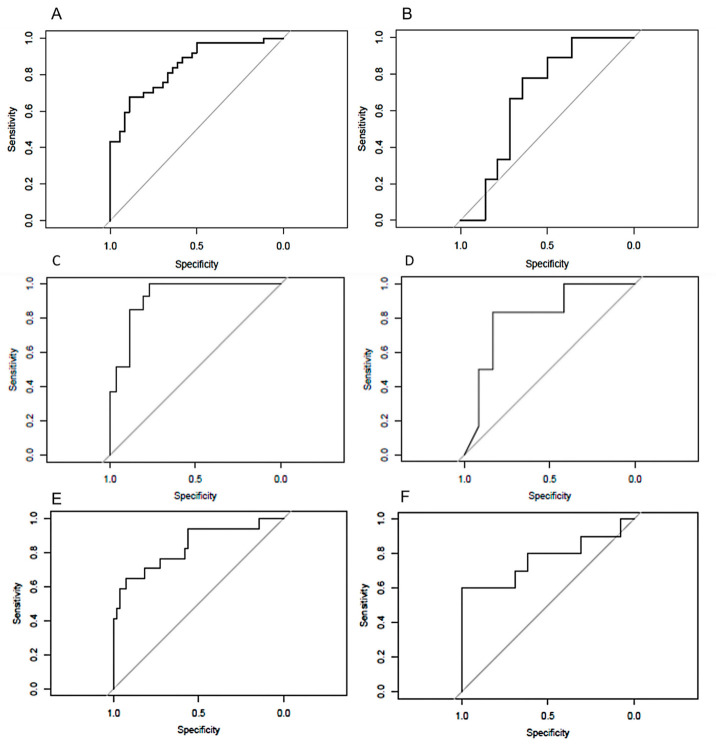
ROC curves of models to predict DMI on the training (**A**) and validation (**B**) sets, to predict LVSI on the training (**C**) and validation (**D**) sets, and to predict low-risk endometrial cancers on the training (**E**) and validation (**F**) sets.

**Table 1 jpm-12-01854-t001:** Acquisition parameters of the MRI sequences employed to observe endometrial tumors.

Parameter	Sequence	Acquisition Plane	Repetition Time/Echo Time (ms)	Matrix	Field of View (cm)	Sections: Number, Thickness (mm), Spacing (mm)	*b* Value (s/mm^2^)
T1-weighted	Spin-echo	Axial	470/3	448 × 288	24	30; 4; 0.5	
T2-weighted	Fast Recovery Fast spin echo	Axial, sagittal, oblique axial *	4000–4500/85	384 × 256	24	30; 4; 0.5	
Diffusion Weighted Imaging	Echo-planar imaging	Sagittal, oblique axial *	5000/69	128 × 128	28	30; 4; 0.5	0, 800
DCE T1-weighted imaging †	Three-dimensional gradient recalled echo	Sagittal, oblique axial *	7/2	320 × 224	28	128; 3;	

* The oblique axial plane was perpendicular to the endometrial cavity, resulting in a short-axis view. † DCE imaging was performed after administration of 0.1 mmol/kg of body weight of gadolinium chelate [ProHance (Gadoteridol)]. Images were acquired at multiple phases of contrast enhancement in sagittal and oblique axial planes (the protocol always includes precontrast sagittal and axial oblique and postcontrast at 80 s in the sagittal plane and 180 s in the oblique axial plane).

**Table 2 jpm-12-01854-t002:** Clinical and histological characteristics for patients with endometrial cancer, included in the training set and validation set.

Training Set		Validation Set	
*n*. patients	98	*n*. patients	26
Age—years (mean)	62	Age—years (mean)	58
Median tumor diameter (mm) (range)	36 (3–95 mm)	Median tumor diameter (mm) (range)	28 (15–60 mm)
Grading:		Grading:	
G1	9	G1	11
G2	50	G2	10
G3	38	G3	5
Not available (NA)	1	Not available (NA)	0
Histology:		Histology:	
Endometrioid	79	Endometrioid	25
Non-endometrioid (serous/clear cell)	19	Non-endometrioid (serous/clear cell)	1
Myometrial invasion:		Myometrial invasion:	
<50%	48	<50%	16
≥50%	50	≥50%	10
Tumor diameter:		Tumor diameter:	
<2 cm	86	<2 cm	6
2 cm	11	2 cm	17
NA	1	NA	4
LVSI:		LVSI:	
No	44	No	14
Yes	43	Yes	6
NA	11	NA	6
Cervical stromal invasion:		Cervical stromal invasion:	
No	71	No	20
Yes	27	Yes	6
Nodal metastases:		Nodal metastases:	
No	52	No	22
Yes	22	Yes	4
NA	24		
Adnexal involvement:		Adnexal involvement:	
No	88	No	22
Yes	10	Yes	4
Vaginal/parametrial involvement:		Vaginal/parametrial involvement:	
No	94	No	24
Yes	4	Yes	2
FIGO staging:		FIGO staging:	
IA	40	IA	13
IB	20	IB	3
II	8	II	2
IIIA	4	IIIA	1
IIIB	4	IIIB	1
IIIC	20	IIIC	5
IVA	0	IVA	0
IVB	2	IVB	1

**Table 3 jpm-12-01854-t003:** Performance and Accuracy metrics of Radiomics Models in Training and Validation sets for predicting DMI and LVSI and discriminating low-risk group versus the other risk groups.

Radiomics Models	Training Set						Validation Set					
	Sensitivity (%)	Specificity (%)	PPV (%)	NPV (%)	Accuracy (95% CI)	AUC	Sensitivity (%)	Specificity (%)	PPV (%)	NPV (%)	Accuracy (95% CI)	AUC
**DMI prediction**	0.67	0.89	0.86	0.72	0.78 (0.67–0.86)	0.85	0.66	0.71	0.60	0.76	0.69 (0.47–0.86)	0.68
**LVSI prediction**	1.00	0.77	0.81	1.00	0.89 (0.77–0.95)	0.92	0.83	0.83	0.71	0.90	0.83 (0.58–0.96)	0.81
**Low-risk discrimination**	0.64	0.93	0.73	0.89	0.86 (0.76–0.93	0.84	0.60	1.00	1.00	0.76	0.82 (0.61–0.95)	0.76

**Table 4 jpm-12-01854-t004:** Main information about the articles discussed and compared for assessment of radiomics in endometrial cancer.

Authors	Year	Study Design	Number of Patients	Imaging Technique	Software	Main Conclusions
Ueno et al. [12]	2017	Retrospective	137	T2-WI, diffusion-weighted imaging (DWI), Apparent diffusion coefficient (ADC) and T1-W post contrast images	TexRAD	Texture features (TF) associated with DMI, LVSI, and high-grade tumor
Stanzione et al. [26]	2020	Retrospective	54	T2-WI	PyRadiomics	Radiomics model increased radiologist performance for DMI detection
Y. Han et al. [24]	2020		163	T2-WI and DWI	PyRadiomics	Whole-uterine MRI radiomic features show potential in predicting DMI
Ytre-Huage et al. [23]	2018	Prospective	180	ADC	TexRAD	TF independently predicted DMI, high-risk histological subtype and reduced survival
Fasmer et al. [25]	2021	Retrospective	138	T1-W post-contrast images	Python	Medium-to-high AUCs for prediction of DMI, lymphnode (LN) metastasis, FIGO stage, and poor outcome
Xu et al. [28]	2019	Retrospective	200	T2-WI and T1-W post-contrast images	Python	Model based on radiomic and clinical features showed good discrimination of positive LN, especially for normal-sized LN
Bereby-Kahane et al. [29]	2020	Retrospective	73	T2-WI and ADC	TexRAD	TF is of limited value to predict high grade and LVSI
Yan et al. [30]	2021	Retrospective	622	T2-WI, DWI, ADC, and T1-W post contrast images	Pyradiomics	Higher diagnostic performance for radiomics model than for radiologists alone to assess pelvic LN status
Yan et al. [13]	2020	Retrospective	717	T2-WI, DWI, ADC, and T1-W post contrast images	Pyradiomics	Radiomics nomogram shows good performance in risk prediction
T. L. Lefebvre et al. [31]	2022	Retrospective	157	T2-WI, DWI, and T1-W post contrast images	Pyradiomics	Three-dimensional radiomics stratify patients according to FIGO stage, high grade, DMI, LVSI
P.P. Mainenti et al. [32]	2022	Retrospective	133	T2-WI	PyRadiomics	Whole-lesion radiomics showed encouraging results for the identification of low-risk patients
D. Liu et al. [33]	2022	Retrospective	202	T2WI, ADC and T1-W post contrast images	PyRadiomics	Model incorporating clinical and radiomic findings predict 5-year survival

## Data Availability

The data presented in this study are available on request from the corresponding author.

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
