# Peer review of "Preoperative Tumor Texture Analysis on MRI for High-Risk Disease Prediction in Endometrial Cancer: A Hypothesis-Generating Study"

_jpm, 2022, doi:10.3390/jpm12111854_

Round 1

Reviewer 1 Report

The study aimed to develop/validate MRI radiomics models for high-risk endometrial cancer (EC) prediction preoperatively. Women with surgically treated EC between April 2009 and May 2019 were analyzed (N=73 in center 1 as training dataset and N=23 in center 2 as validation dataset). Tumor segmentations were manually delineated and 228 radiomics features were calculated. Radiomic models yielded AUCT/AUCV of 0.84/0.76 for differentiating low-risk vs other risk-classes.

 1. The numbers of both training and validation datasets are too small for this kind of radiomic study and the significant features are probably false positive or correlated with other factors such as tumor size.

 2. The study of radiomics is complicated by multiple issues. One of the most important issues is that multiple hypothesis testing results in significant inflation of type I errors and can easily lead to false-positive findings. The number of calculated features and the number of possible variations in parameters should all be adjusted for statistical analyses. If you use combination of features for modeling, the multiple comparisons of possible combinations also need to be adjusted.

 3. As mentioned by authors as a limitation, data were collected over a long time (2009-2019) and MRI technology has improved with more sophisticated machinery and better image quality. Are the machines and imaging protocols the same between the two centers for all included patients?

 4. Table 1 and Supplementary Table 1 can be combined into one table.

 5. Nearly all the references numbers in Table 3 were incorrect, such as Ueno et al. is ref. #10, not ref. #6, Stanzione et al is ref 22, not ref. #17, Ytre-Hauge et al is Ref. #20, not ref. #16, … Besides, Xu et al 2019 and Yan et al 2021 were not listed in the references.

Author Response

The study aimed to develop/validate MRI radiomics models for high-risk endometrial cancer (EC) prediction preoperatively. Women with surgically treated EC between April 2009 and May 2019 were analyzed (N=73 in center 1 as training dataset and N=23 in center 2 as validation dataset). Tumor segmentations were manually delineated and 228 radiomics features were calculated. Radiomic models yielded AUCT/AUCV of 0.84/0.76 for differentiating low-risk vs other risk-classes.

  1. The numbers of both training and validation datasets are too small for this kind of radiomic study and the significant features are probably false positive or correlated with other factors such as tumor size.

 → We understand the reviewer’s concern, regarding the numerosity of our study population. We have generally toned down the manuscript highlighting statistical associations between radiomic features and DMI, lymphovascular space invasion (LVSI), and low-risk category rather than making concrete claims of clinical usefulness. We acknowledge our preliminary could vary and require further validation. The conclusion now reads: MRI-based radiomics have great potential in developing advanced prognostication in EC. The title has been reworded as follows: Preoperative Tumor Texture Analysis on MRI for High-Risk disease prediction in Endometrial Cancer: a hypothesis generating study.

 The study of radiomics is complicated by multiple issues. One of the most important issues is that multiple hypothesis testing results in significant inflation of type I errors and can easily lead to false-positive findings. The number of calculated features and the number of possible variations in parameters should all be adjusted for statistical analyses. If you use combination of features for modeling, the multiple comparisons of possible combinations also need to be adjusted.

We thank the reviewer for pointing this out. We understand the reviewer’s concern that our study may be underpowered. We acknowledge that our preliminary results could require further and larger validation. Nevertheless, we strictly applied the TRIPOD criteria, pursuing an external validation (TRIPOD 3B), making the need of multiple testing less overriding.

Multiple testing techniques have anyway been applied, as described in line 185-187.

These issues have been added in the limitation section, in line with reviewers’ comments regarding the need to tone down the manuscript to reflect this: The number of patients was relatively low, even if it allowed for a dataset for further model validation. This also limited the statistical power of testing to assess differences in performance.

 As mentioned by authors as a limitation, data were collected over a long time (2009-2019) and MRI technology has improved with more sophisticated machinery and better image quality. Are the machines and imaging protocols the same between the two centers for all included patients?

→ Yes. This has now been better specified in MRI Imaging section of the Materials and Methods. The following was added: The pelvic MR imaging protocol in the training and validation sets included T2-weighted imaging (WI) in axial, sagittal, axial oblique plane, Diffusion Weighted Imaging (DWI), T1-W post-contrast images.

  1. Table 1 and Supplementary Table 1 can be combined into one table.

→ Done

  1. 5. Nearly all the references numbers in Table 3 were incorrect, such as Ueno et al. is ref. #10, not ref. #6, Stanzione et al is ref 22, not ref. #17, Ytre-Hauge et al is Ref. #20, not ref. #16, … Besides, Xu et al 2019 and Yan et al 2021 were not listed in the references.

→ Done

Reviewer 2 Report

The paper is very interesting and falls within the scope of the journal.

Here my concers:
- erase multicenter (only 2 centers are included...)

- take into account these papers to improve your introduction and reference list on endometrial cancer (doi: 10.1016/j.ygyno.2021.03.029; doi: 10.1016/j.ygyno.2020.05.012)

I have appreciated the well described statistical analysis.

Results are well written.

Conclusion are in line what has been reported in main text

Author Response

The paper is very interesting and falls within the scope of the journal.

  1. Erase multicenter (only 2 centers are included)

→ Done. The text has been reworded as follows: two-center study.

- take into account these papers to improve your introduction and reference list on endometrial cancer (doi: 10.1016/j.ygyno.2021.03.029; doi: 10.1016/j.ygyno.2020.05.012)

→ Done. We thank the reviewer for this helpful advice and have updated our text and references.

I have appreciated the well described statistical analysis.

Results are well written.

Conclusion are in line what has been reported in main text

Reviewer 3 Report

This well-written and interesting work has several strengths (e.g., multicenter, robust radiomics pipeline, clinically driven and clear research question, light workflow based on a single sequence etc.). Nevertheless, there is still room for improvement and some points need to be clarified:

1) In the Introduction, would the Authors consider mentioning this recent systematic review and quality appraisal to provide readers with a more comprehensive overview of the current knowledge in the topic? (10.1016/j.ejso.2021.06.023)

2) Would the Authors consider adding a flow-chart to easily show the patient selection process for the study?

3) Feature stability testing for multiple segmentation was performed (which is a good thing) but the segmentation process details are not completely clear: how were the segmentations needed for testing obtained?

4)  Would the Authors consider adding the acquisition protocol details as supplementary materials?

5) In tables 1 and 2, would the Authors please consider reporting data for the actual instances used for training (n= 73) and testing (n= 23)? Also, would the Authors consider merging the two tables so that comparisons between the characteristics of train and test set could be made at a glance (the Authors might also want to support these with statistical analyses)?

6) Since the same scanner was used in the two institutions, is the external validation properly external? Technical parameters might differ, so I suppose that it still counts, but do the Authors believe this issue deserves to be mentioned among the limitations of the study?

7) It is rather unusual to present a new table in the discussion section but I think an exception could be made in this case since it fits particularly well and helps the readers put the reported results in the context of relevant literature. However, there are some additional publications that should be considered for the sake of comprehensiveness: 10.1148/radiol.212873 and 10.1016/j.ejrad.2022.110226 and 10.3389/fonc.2022.813069 and 10.4103/jcrt.JCRT_1393_20

8)  Figure 2 is a bit confusing in the present form, since the panel includes two As and two Bs; furthermore, I was not able to find the arrows that are mentioned in the figure legend; finally, since the figure depicts the segmentation process, would the Authors consider moving it from the Discussion to the M&M section?

9) Manual segmentation is indeed a tedious and time-consuming task and the Authors correctly acknowledge this in the limitations; however, when mentioning automated segmentation, they might want to consider citing the relevant literature: 10.1038/s41598-020-80068-9 and 10.1038/s41598-021-93792-7

Author Response

This well-written and interesting work has several strengths (e.g., multicenter, robust radiomics pipeline, clinically driven and clear research question, light workflow based on a single sequence etc.). Nevertheless, there is still room for improvement and some points need to be clarified:

1) In the Introduction, would the Authors consider mentioning this recent systematic review and quality appraisal to provide readers with a more comprehensive overview of the current knowledge in the topic? (10.1016/j.ejso.2021.06.023)

 → Done. We thank the reviewer for this helpful advice and have updated our Introduction and references. The following was added: Despite promising performances of this filed in clinical practice, only few studies have explored MRI-based radiomic tumor features in endometrioid EC and linked these to an aggressive phenotype.

2) Would the Authors consider adding a flow-chart to easily show the patient selection process for the study?

 → Done. Figure 1 has been added.

3) Feature stability testing for multiple segmentation was performed (which is a good thing) but the segmentation process details are not completely clear: how were the segmentations needed for testing obtained?

→ The segmentation process has been better reported in Image Analysis section, as follows: The radiologists, who were aware of diagnosis of endometrial cancer but blinded to clinical and pathologic outcomes of patients, manually segmented tumor regions of interest (ROI) for each case. A ROI was manually drawn along the margin of the visible gross tumor on the axial plane T2-WI along tumor boundary on each slice for each tumor. Thus, volumetric region of interest (VOI) of each tumor was segmented. Tumor contour was defined as area of intermediate signal intensity on T2-WI that were different from normal adjacent low signal intensity myometrium. Although not used for segmentation, DWI and T1-WI acquired post-contrast injection series were available to the readers for visual inspection support segmentations in case of need. 

4)  Would the Authors consider adding the acquisition protocol details as supplementary materials?

 → Done. This has now been added as Table 1 suppl.

5) In tables 1 and 2, would the Authors please consider reporting data for the actual instances used for training (n= 73) and testing (n= 23)? Also, would the Authors consider merging the two tables so that comparisons between the characteristics of train and test set could be made at a glance (the Authors might also want to support these with statistical analyses)?

→ We have merges table 1/table1supp (clinical characteristic) in unique table for validation and training sets. Furthermore, we have merged the results for training and validation set in a single table, as suggested by the reviewer. We fully agree the reviewer’s suggestion, however we did not feel additional statistical analyses requested above was warranted.

6) Since the same scanner was used in the two institutions, is the external validation properly external? Technical parameters might differ, so I suppose that it still counts, but do the Authors believe this issue deserves to be mentioned among the limitations of the study?

 We understand the reviewer’s concern. One limit of radiomic study may be represented by the inhomogeneity of the training and validation cohorts in terms of MRI vendors, images, technical characteristics. To understand the applicability of the radiomic model to the general population, future developments of this innovative approach can therefore take into account further external validation cohorts with images acquired on scanners by different vendors, as already done for other pathologies like rectal cancer [Dinapoli N, Barbaro B, Gatta R, et al (2018) Magnetic Resonance, Vendor-independent, Intensity Histogram Analysis Predicting Pathologic Complete Response After Radiochemotherapy of Rectal Cancer. Int J Radiat Oncol Biol Phys 102:765–774. https://doi.org/10.1016/j.ijrobp.2018.04.065]. In line with the reviewer’s suggestions, we have added this issue in the limitation section: Data were collected over a long time (2009- 2019) and MRI technology has improved with more sophisticated machinery, different technical parameters, and better image quality. All these aspects might have impacted on textural radiomic features extraction and subsequent modeling.

7) It is rather unusual to present a new table in the discussion section but I think an exception could be made in this case since it fits particularly well and helps the readers put the reported results in the context of relevant literature. However, there are some additional publications that should be considered for the sake of comprehensiveness: 10.1148/radiol.212873 and 10.1016/j.ejrad.2022.110226 and 10.3389/fonc.2022.813069 and 10.4103/jcrt.JCRT_1393_20

 → Done. We thank the reviewer for this helpful advice and have updated our discussion and references.

8)  Figure 2 is a bit confusing in the present form, since the panel includes two As and two Bs; furthermore, I was not able to find the arrows that are mentioned in the figure legend; finally, since the figure depicts the segmentation process, would the Authors consider moving it from the Discussion to the M&M section?

 → Apologies for the error, the picture has been corrected and moved to the M&M section.

9) Manual segmentation is indeed a tedious and time-consuming task and the Authors correctly acknowledge this in the limitations; however, when mentioning automated segmentation, they might want to consider citing the relevant literature: 10.1038/s41598-020-80068-9 and 10.1038/s41598-021-93792-7

→ Done. We thank the reviewer for this helpful advice and have updated our Introduction and References.

Round 2

Reviewer 3 Report

The Authors satisfactorily addressed all my comments further improving the quality of their manuscript. I do not have additional remarks and endorse the publication of this work.